# MULTI-RESOLUTION CONTINUOUS NORMALIZING FLOWS

## ABSTRACT

Recent work has shown that Neural Ordinary Differential Equations (ODEs) can serve as generative models of images using the perspective of Continuous Normalizing Flows (CNFs). Such models offer exact likelihood calculation, and invertible generation/density estimation. In this work we introduce a Multi-Resolution variant of such models (MRCNF), by characterizing the conditional distribution over the additional information required to generate a fine image that is consistent with the coarse image. We introduce a transformation between resolutions that allows for no change in the log likelihood. We show that this approach yields comparable likelihood values for various image datasets, using orders of magnitude fewer parameters than the prior methods, in significantly less training time, using only one GPU.

## 1 INTRODUCTION

Reversible generative models derived through the use of the change of variables technique (Dinh et al., 2017; Kingma & Dhariwal, 2018; Ho et al., 2019a; Yu et al., 2020) are growing in interest as alternatives to generative models based on Generative Adversarial Networks (GANs) (Goodfellow et al., 2016) and Variational Autoencoders (VAEs) (Kingma & Welling, 2013). While GANs and VAEs have been able to produce visually impressive samples of images, they have a number of limitations. A change of variables approach facilitates the transformation of a simple base probability distribution into a more complex model distribution. Reversible generative models using this technique are attractive because they enable efficient density estimation, efficient sampling, and computation of exact likelihoods.

A promising variation of the change-of-variable approach is based on the use of a continuous time variant of normalizing flows (Chen et al., 2018a; Grathwohl et al., 2019; Finlay et al., 2020), which uses an integral over continuous time dynamics to transform a base distribution into the model distribution, called Continuous Normalizing Flows (CNF). This approach uses ordinary differential equations (ODEs) specified by a neural network, or Neural ODEs. CNFs have been shown to be capable of modelling complex distributions such as those associated with images.

While this new paradigm for the generative modelling of images is not as mature as GANs or VAEs in terms of the generated image quality, it is a promising direction of research as it does not have some key shortcomings associated with GANs and VAEs. Specifically, GANs are known to suffer from mode-collapse (Lin et al., 2018), and are notoriously difficult to train (Arjovsky & Bottou, 2017) compared to feed forward networks because their adversarial loss seeks a saddle point instead of a local minimum (Berard et al., 2020). CNFs are trained by mapping images to noise, and their reversible architecture allows images to be generated by going in reverse, from noise to images. This leads to fewer issues related to mode collapse, since any input example in the dataset can be recovered from the flow using the reverse of the transformation learned during training. VAEs only provide a lower bound on the marginal likelihood whereas CNFs provide exact likelihoods. Despite the many advantages of reversible generative models built with CNFs, quantitatively such methods still do not match the widely used Fréchet Inception Distance (FID) scores of GANs or VAEs. However their other advantages motivate us to explore them further.

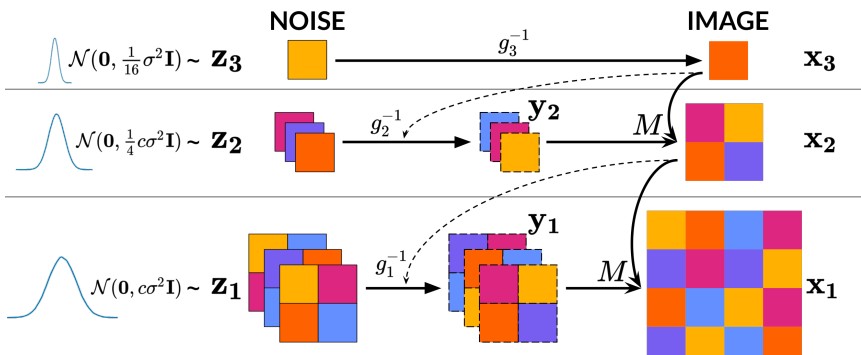

Figure 1: The architecture of our Multi-Resolution Continuous Normalizing Flow (MRCNF) method (best viewed in color). Continuous normalizing flows (CNFs) $g_s$ are used to generate images $\mathbf{x}_s$ from noise $\mathbf{z}_s$ at each resolution, with those at finer resolutions conditioned (dashed lines) on the coarser image one level above $\mathbf{x}_{s+1}$, except at the coarsest level where it is unconditional. Every finer CNF produces an intermediate image $\mathbf{y}_s$, which is then combined with the immediate coarser image $\mathbf{x}_{s+1}$ using a linear map $M$ to form $\mathbf{x}_s$.

Furthermore, state-of-the art GANs and VAEs exploit the multi-resolution properties of images, and recent top-performing methods also inject noise at each resolution (Brock et al., 2019; Shaham et al., 2019; Karras et al., 2020; Vahdat & Kautz, 2020). While shaping noise is fundamental to normalizing flows, only recently have normalizing flows exploited the multi-resolution properties of images. For example, WaveletFlow (Yu et al., 2020) splits an image into multiple resolutions using the Discrete Wavelet Transform, and models the average image at each resolution using a normalizing flow. While this method has advantages, it suffers from many issues such as high parameter count and long training time.

In this work, we consider a non-trivial multi-resolution approach to continuous normalizing flows, which fixes many of these issues. A high-level view of our approach is shown in Figure 1. Our main contributions are:

1. We propose a multi-resolution transformation that does not add cost in terms of likelihood.

2. We introduce **Multi-Resolution Continuous Normalizing Flows (MRCNF)**.

3. We achieve comparable Bits-per-dimension (BPD) (negative log likelihood per pixel) on image datasets using fewer model parameters and significantly less training time with only one GPU.

## 2 BACKGROUND

### 2.1 NORMALIZING FLOWS

Normalizing flows (Tabak & Turner, 2013; Jimenez Rezende & Mohamed, 2015; Dinh et al., 2017; Papamakarios et al., 2019; Kobyzev et al., 2020) are generative models that map a complex data distribution, such as real images, to a known noise distribution. They are trained by maximizing the log likelihood of their input images. Suppose a normalizing flow $g$ produces output $\mathbf{z}$ from an input $\mathbf{x}$ i.e. $\mathbf{z} = g(\mathbf{x})$. The change-of-variables formula provides the likelihood of the image under this transformation as:

$$\log p(\mathbf{x}) = \log \left| \det \frac{\mathrm{d}g}{\mathrm{d}\mathbf{x}} \right| + \log p(\mathbf{z}) \qquad (1)$$

The first term on the right (log determinant of the Jacobian) is often intractable, however, previous works on normalizing flows have found ways to estimate this efficiently. The second term, $\log p(\mathbf{z})$, is computed as the log probability of $\mathbf{z}$ under a known noise distribution, typically the standard Gaussian $\mathcal{N}(\mathbf{0}, \mathbf{I})$.

## 2.2 WAVELET FLOW (YU ET AL., 2020)

WaveletFlow decomposes an image using the Discrete Wavelet Transformation, and maps the average image at each resolution to noise using a normalizing flow. The WaveletFlow model builds on the Glow (Kingma & Dhariwal, 2018) architecture, with a high parameter count and a long training time. It uses an orthogonal transformation, which does not preserve the range of the image, and adds a constant term to the log likelihood at each resolution. We aim to fix these issues using our MRCNF.

## 2.3 CONTINUOUS NORMALIZING FLOWS

Continuous Normalizing Flows (CNF) (Chen et al., 2018a; Grathwohl et al., 2019; Finlay et al., 2020) are a variant of normalizing flows that operate in the continuous domain. A CNF creates a geometric flow between the input and target (noise) distributions, by assuming that the state transition is governed by an Ordinary Differential Equation (ODE). It further assumes that the differential function is parameterized by a neural network, this model is called a Neural ODE (Chen et al., 2018a). Suppose CNF $g$ transforms its state $\mathbf{v}(t)$ using a Neural ODE, with neural network $f$ defining the differential. Here, $\mathbf{v}(t_0) = \mathbf{x}$ is, say, an image, and at the final time step $\mathbf{v}(t_1) = \mathbf{z}$ is a sample from a known noise distribution.

$$\frac{\mathrm{d}\mathbf{v}(t)}{\mathrm{d}t} = f(\mathbf{v}(t), t) \implies \mathbf{v}(t_1) = g(\mathbf{v}(t_0)) = \mathbf{v}(t_0) + \int_{t_0}^{t_1} f(\mathbf{v}(t), t)\,\mathrm{d}t \tag{2}$$

This integration is typically performed by an ODE solver. Since this integration can be run backwards as well to obtain the same $\mathbf{v}(t_0)$ from $\mathbf{v}(t_1)$, a CNF is a reversible model.

Equation 1 can be used to compute the change in log-probability induced by the CNF. However, Chen et al. (2018a) and Grathwohl et al. (2019) proposed a more efficient variant in the context of CNFs, called the instantaneous change-of-variables formula:

$$\frac{\partial \log p(\mathbf{v}(t))}{\partial t} = -\mathrm{Tr}\left(\frac{\partial f}{\partial \mathbf{v}(t)}\right) \implies \Delta \log p_{\mathbf{v}(t_0) \to \mathbf{v}(t_1)} = -\int_{t_0}^{t_1} \mathrm{Tr}\left(\frac{\partial f}{\partial \mathbf{v}(t)}\right)\mathrm{d}t \tag{3}$$

Hence, the change in log-probability of the state of the Neural ODE i.e. $\Delta \log p_{\mathbf{v}}$ is expressed as another differential equation. The ODE solver now solves both differential equations eq. (2) and eq. (3) by augmenting the original state with the above. Thus, a CNF provides both the final state $\mathbf{v}(t_1)$ as well as the change in log probability $\Delta \log p_{\mathbf{v}(t_0) \to \mathbf{v}(t_1)}$ together.

Prior works (Grathwohl et al., 2019; Finlay et al., 2020; Ghosh et al., 2020; Onken et al., 2021; Huang & Yeh, 2021) have trained CNFs as reversible generative models of images, by maximizing the likelihood of the images under the model:

$$\mathbf{z} = g(\mathbf{x}) \quad ; \qquad \log p(\mathbf{x}) = \Delta \log p_{\mathbf{x} \to \mathbf{z}} + \log p(\mathbf{z}) \tag{4}$$

where $\mathbf{x}$ is an image, $\mathbf{z}$ and $\Delta \log p_{\mathbf{x} \to \mathbf{z}}$ are computed by the CNF using eq. (2) and eq. (3), and $\log p(\mathbf{z})$ is the likelihood of the computed $\mathbf{z}$ under a known noise distribution, typically the standard Gaussian $\mathcal{N}(\mathbf{0}, \mathbf{I})$. Novel images are generated by sampling $\mathbf{z}$ from the known noise distribution, and running it through the CNF in reverse.

## 3 OUR METHOD

Our method is a reversible generative model of images that builds on top of CNFs. We introduce the notion of multiple resolutions in images, and connect the different resolutions in an autoregressive fashion. This helps generate images faster, with better likelihood values at higher resolutions, using only one GPU in all our experiments. We call this model Multi-Resolution Continuous Normalizing Flow (MRCNF).

### 3.1 MULTI-RESOLUTION IMAGE REPRESENTATION

Multi-resolution representations of images have been explored in computer vision for decades (Burt, 1981; Marr, 2010; Witkin, 1987; Burt & Adelson, 1983; Mallat, 1989; Lindeberg, 1990). This implies that much of the content of an image at a resolution is a composition of low-level information captured at coarser resolutions, and some high-level information not present in the coarser images. We take advantage of this property by first decomposing an image in *resolution space* i.e. by expressing it as a series of $S$ images at decreasing resolutions: $\mathbf{x} \rightarrow (\mathbf{x}_1, \mathbf{x}_2, \ldots, \mathbf{x}_S)$, where $\mathbf{x}_1 = \mathbf{x}$ is the finest image, $\mathbf{x}_S$ is the coarsest, and every $\mathbf{x}_{s+1}$ is the average image of $\mathbf{x}_s$. This called an image pyramid, or a Gaussian Pyramid if the upsampling-downsampling operations include a Gaussian filter (Burt, 1981; Burt & Adelson, 1983; Adelson et al., 1984; Witkin, 1987; Lindeberg, 1990). In this work, we obtain a coarser image simply by averaging pixels in every 2×2 patch, thereby halving the width and height.

However, this representation is redundant since much of the information in $\mathbf{x}_1$ is contained in $\mathbf{x}_{s>1}$. Instead, we express $\mathbf{x}$ as a series of high-level information $\mathbf{y}_s$ not present in the immediate coarser images $\mathbf{x}_{s+1}$, and a final coarse image $\mathbf{x}_S$:

$$\mathbf{x} \rightarrow (\mathbf{y}_1, \mathbf{x}_2) \rightarrow (\mathbf{y}_1, \mathbf{y}_2, \mathbf{x}_3) \rightarrow \cdots \rightarrow (\mathbf{y}_1, \mathbf{y}_2, \ldots, \mathbf{y}_{S-1}, \mathbf{x}_S) \tag{5}$$

Our overall method is to map these $S$ terms to $S$ noise samples using $S$ CNFs.

### 3.2 DEFINING THE HIGH-LEVEL INFORMATION $\mathbf{y}_s$

We choose to design a linear transformation with the following properties: 1) invertible i.e. it should be possible to deterministically obtain $\mathbf{x}_s$ from $\mathbf{y}_s$ and $\mathbf{x}_{s+1}$, and vice versa ; 2) volume preserving i.e. determinant is 1, change in log-likelihood is 0 ; 3) angle preserving ; and 4) range preserving (under the notion of the maximum principle (Varga, 1966)).

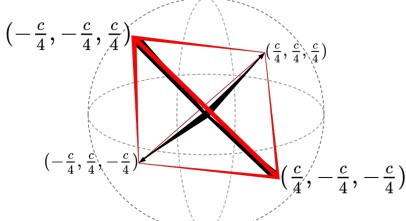

Figure 2: Tetrahedron in 3D space with 4 corners

Consider the simplest case of 2 resolutions where $\mathbf{x}_1$ is a 2×2 image with pixel values $x_1, x_2, x_3, x_4$, and $\mathbf{x}_2$ is a 1×1 image with pixel value $\bar{x} = \frac{1}{4}(x_1 + x_2 + x_3 + x_4)$. We require three values $(y_1, y_2, y_3) = \mathbf{y}_1$ that contain information not present in $\mathbf{x}_2$, such that $\mathbf{x}_1$ is obtained when $\mathbf{y}_1$ and $\mathbf{x}_2$ are combined.

This could be viewed as a problem of finding a matrix $\mathbf{M}$ such that: $[x_1, x_2, x_3, x_4]^\top = \mathbf{M}[y_1, y_2, y_3, \bar{x}]^\top$. We fix the last column of $\mathbf{M}$ as $[1, 1, 1, 1]^\top$, since every pixel value in $\mathbf{x}_1$ depends on $\bar{x}$. Finding the rest of the parameters can be viewed as requiring four 3D vectors that are spaced such that they do not degenerate the number of dimensions of their span. These can be considered as the four corners of a tetrahedron in 3D space, under any configuration (rotated in 3D space), and any scaling of the vectors (see Figure 2).

Out of the many possibilities for this tetrahedron, we could choose the matrix that performs the Discrete Haar Wavelet Transform (Mallat, 1989; Mallat & Peyré, 2009):

$$\begin{bmatrix} x_1 \\ x_2 \\ x_3 \\ x_4 \end{bmatrix} = \begin{bmatrix} \frac{1}{2} & \frac{1}{2} & \frac{1}{2} & 1 \\ \frac{1}{2} & -\frac{1}{2} & -\frac{1}{2} & 1 \\ -\frac{1}{2} & \frac{1}{2} & -\frac{1}{2} & 1 \\ -\frac{1}{2} & -\frac{1}{2} & \frac{1}{2} & 1 \end{bmatrix} \begin{bmatrix} y_1 \\ y_2 \\ y_3 \\ \bar{x} \end{bmatrix} \iff \begin{bmatrix} y_1 \\ y_2 \\ y_3 \\ \bar{x} \end{bmatrix} = \begin{bmatrix} \frac{1}{2} & \frac{1}{2} & -\frac{1}{2} & -\frac{1}{2} \\ \frac{1}{2} & -\frac{1}{2} & \frac{1}{2} & -\frac{1}{2} \\ \frac{1}{2} & -\frac{1}{2} & -\frac{1}{2} & \frac{1}{2} \\ \frac{1}{4} & \frac{1}{4} & \frac{1}{4} & \frac{1}{4} \end{bmatrix} \begin{bmatrix} x_1 \\ x_2 \\ x_3 \\ x_4 \end{bmatrix} \tag{6}$$

However, this has $\log \left| \det(\mathbf{M}^{-1}) \right| = \log(1/2)$ (eq. (6)), and is therefore not volume preserving. Other simple scaling of eq. (6) has been used in the past, for example multiplying the last row of eq. (6) by 2, yielding an orthogonal transformation, such as in WaveletFlow (Yu et al., 2020). However, this transformation neither preserves the volume i.e. the log determinant is not 0, nor the maximum i.e. the range of $\mathbf{x}_s$ changes.

We wish to find a transformation $\mathbf{M}$ where: one of the results is the average of the inputs, $\bar{x}$; it is unit determinant; the columns are orthogonal; and it preserves the range of $\bar{x}$. Fortunately such a matrix exists – although we have not seen it discussed in prior literature. It can be seen as a variant of the Discrete Haar Wavelet Transformation matrix that is unimodular, i.e. has a determinant of 1 (and is therefore volume preserving), while also preserving the range of the images for the input and its average:

$$\begin{bmatrix} x_1 \\ x_2 \\ x_3 \\ x_4 \end{bmatrix} = \frac{1}{a} \begin{bmatrix} c & c & c & a \\ c & -c & -c & a \\ -c & c & -c & a \\ -c & -c & c & a \end{bmatrix} \begin{bmatrix} y_1 \\ y_2 \\ y_3 \\ \bar{x} \end{bmatrix} \iff \begin{bmatrix} y_1 \\ y_2 \\ y_3 \\ \bar{x} \end{bmatrix} = \begin{bmatrix} c^{-1} & c^{-1} & -c^{-1} & -c^{-1} \\ c^{-1} & -c^{-1} & c^{-1} & -c^{-1} \\ c^{-1} & -c^{-1} & -c^{-1} & c^{-1} \\ a^{-1} & a^{-1} & a^{-1} & a^{-1} \end{bmatrix} \begin{bmatrix} x_1 \\ x_2 \\ x_3 \\ x_4 \end{bmatrix} \tag{7}$$

where $c = 2^{2/3}$, $a = 4$. Hence, $\log \left| \det(\mathbf{M}^{-1}) \right| = \log(1) = 0$. This can be scaled up to larger spatial regions by performing the same calculation for each 2×2 patch. Let $M$ be the function that uses matrix $\mathbf{M}$ from above and combines every pixel in $\mathbf{x}_{s+1}$ with the three corresponding pixels in $\mathbf{y}_s$ to make the 2×2 patch at that location in $\mathbf{x}_s$ using eq. (7):

$$\mathbf{x}_s = M(\mathbf{y}_s, \mathbf{x}_{s+1}) \iff \mathbf{y}_s, \mathbf{x}_{s+1} = M^{-1}(\mathbf{x}_s) \tag{8}$$

Equation 1 can be used to compute the change in log likelihood from this transformation $\mathbf{x}_s \to (\mathbf{y}_s, \mathbf{x}_{s+1})$:

$$\log p(\mathbf{x}_s) = \Delta \log p_{\mathbf{x}_s \to (\mathbf{y}_s, \mathbf{x}_{s+1})} + \log p(\mathbf{y}_s, \mathbf{x}_{s+1})$$
$$= \log \left| \det(M^{-1}) \right| + \log p(\mathbf{y}_s \mid \mathbf{x}_{s+1}) + \log p(\mathbf{x}_{s+1}) \tag{9}$$

where $\log \left| \det(M^{-1}) \right| = \text{dims}(\mathbf{x}_{s+1}) \log(1/2)$ in the case of eq. (6), where "dims" is the number of pixels times the number of channels (typically 3) in the image, and $\log \left| \det(M^{-1}) \right| = 0$ for eq. (7). Although a model could learn to compensate for the additional log-likelihood due to eq. (6), we find that using eq. (7) improves log likelihood values and decreases training time.

### 3.3 MULTI-RESOLUTION CONTINUOUS NORMALIZING FLOWS

Using the multi-resolution image representation in eq. (5), we characterize the conditional distribution over the additional information ($\mathbf{y}_s$) required to generate a higher resolution image ($\mathbf{x}_s$) that is consistent with the average ($\mathbf{x}_{s+1}$) over the pixel space. This framework ensures that one coarse image could generate several potential fine images, but these fine images have the same coarse image as their average. This fact is preserved across resolutions. Note that the 3 additional pixels in $\mathbf{y}_s$ per pixel in $\mathbf{x}_{s+1}$ are generated conditioned on the entire coarser image $\mathbf{x}_{s+1}$, thus maintaining consistency using the full context.

At each resolution $s$, we use a CNF to reversibly map between $\mathbf{y}_s$ (or $\mathbf{x}_S$ when $s = S$) and a sample $\mathbf{z}_s$ from a known noise distribution.

In principle, any generative model could be used to map between the multi-resolution image and noise. Normalizing flows are good candidates for this as they are probabilistic generative models that perform exact likelihood estimates, and can be run in reverse to generate novel data from the model's distribution. This allows model comparison and measurement of generalization to unseen data. We choose the CNF variant of normalizing flows at each resolution. CNFs have recently been shown to be effective in modeling image distributions using a fraction of the number of parameters typically used in normalizing flows (and non flow-based approaches), and their underlying framework of Neural ODEs have been shown to be more robust than convolutional layers (Yan et al., 2020).

**Training**: We train an MRCNF by maximizing the average log-likelihood of the images in the training dataset under the model. The log probability of each image $\log p(\mathbf{x})$ can be estimated recursively from eq. (9) as:

$$\log p(\mathbf{x}) = \Delta \log p_{\mathbf{x}_1 \to (\mathbf{y}_1, \mathbf{x}_2)} + \log p(\mathbf{y}_1, \mathbf{x}_2) = \Delta \log p_{\mathbf{x}_1 \to (\mathbf{y}_1, \mathbf{x}_2)} + \log p(\mathbf{y}_1 \mid \mathbf{x}_2) + \log p(\mathbf{x}_2)$$
$$= \sum_{s=1}^{S-1} \left( \Delta \log p_{\mathbf{x}_s \to (\mathbf{y}_s, \mathbf{x}_{s+1})} + \log p(\mathbf{y}_s \mid \mathbf{x}_{s+1}) \right) + \log p(\mathbf{x}_S) \tag{10}$$

where $\Delta \log p_{\mathbf{x}_s \to (\mathbf{y}_s, \mathbf{x}_{s+1})}$ is $\log \left| \det(M^{-1}) \right|$, $\log p(\mathbf{y}_s \mid \mathbf{x}_{s+1})$ and $\log p(\mathbf{x}_S)$ are given by eq. (4):

$$\mathbf{z}_s = g_s(\mathbf{y}_s \mid \mathbf{x}_{s+1}) \quad ; \qquad \log p(\mathbf{y}_s \mid \mathbf{x}_{s+1}) = \Delta \log p_{(\mathbf{y}_s \to \mathbf{z}_s) \mid \mathbf{x}_{s+1}} + \log p(\mathbf{z}_s) \tag{11}$$

$$\mathbf{z}_S = g_S(\mathbf{x}_S) \quad ; \qquad \log p(\mathbf{x}_S) = \Delta \log p_{\mathbf{x}_S \to \mathbf{z}_S} + \log p(\mathbf{z}_S) \tag{12}$$

The coarsest resolution $S$ can be chosen such that the last CNF operates on the image distribution at a small enough resolution that is easy to model unconditionally. All other CNFs are conditioned on the immediate coarser image. The conditioning itself is achieved by concatenating the input image of the CNF with the coarser image. This model could be seen as a stack of CNFs connected in an autoregressive fashion.

Typically, likelihood-based generative models are compared using the metric of bits-per-dimension (BPD), i.e. the negative log likelihood per pixel in the image. Hence, we train our MRCNF to minimize the average BPD of the images in the training dataset, computed using eq. (13):

$$\text{BPD}(\mathbf{x}) = -\log p(\mathbf{x})/\text{dims}(\mathbf{x}) \tag{13}$$

We use FFJORD (Grathwohl et al., 2019) as the baseline model for our CNFs. In addition, we use to two regularization terms introduced by RNODE (Finlay et al., 2020) to speed up the training of FFJORD models by stabilizing the learnt dynamics: the kinetic energy of the flow $\mathcal{K}(\theta)$, and the Jacobian norm $\mathcal{B}(\theta)$:

$$\mathcal{K}(\theta) = \int_{t_0}^{t_1} \|f(\mathbf{v}(t), t, \theta)\|_2^2 \, \mathrm{d}t; \qquad \mathcal{B}(\theta) = \int_{t_0}^{t_1} \|\epsilon^\top \nabla_z f(\mathbf{v}(t), t, \theta)\|_2^2 \, \mathrm{d}t, \quad \epsilon \sim \mathcal{N}(0, I) \tag{14}$$

**Parallel training:** Note that although the final log likelihood $\log p(\mathbf{x})$ involves sequentially summing over values returned by all $S$ CNFs, the log likelihood term of each CNF is independent of the others. Conditioning is done using ground truth images. Hence, each CNF can be trained independently, in parallel.

**Generation**: Given an $S$-resolution model, we first sample $\mathbf{z}_s, s = 1, \ldots, S$ from the latent noise distributions. The CNF $g_s$ at resolution $s$ transforms the noise sample $\mathbf{z}_s$ to high-level information $\mathbf{y}_s$ conditioned on the immediate coarse image $\mathbf{x}_{s+1}$ (except $g_S$ which is unconditioned). $\mathbf{y}_s$ and $\mathbf{x}_{s+1}$ are then combined to form $\mathbf{x}_s$ using $M$ from eq. (7). This process is repeated progressively from coarser to finer resolutions, until the finest resolution image $\mathbf{x}_1$ is computed (see Figure 1). It is to be noted that the generated image at one resolution is used to condition the CNF at the finer resolution.

$$\begin{cases} \mathbf{x}_S = g_S^{-1}(\mathbf{z}_S) & s = S \\ \mathbf{y}_s = g_s^{-1}(\mathbf{z}_s \mid \mathbf{x}_{s+1}); \quad \mathbf{x}_s = M(\mathbf{y}_s, \mathbf{x}_{s+1}) & s = \text{S-1} \to 1 \end{cases} \tag{15}$$

### 3.4 Multi-Resolution Noise

We further decompose the noise image as well into its respective coarser components. This means that ultimately we use only one noise image at the finest level, but it is decomposed into multiple resolutions using eq. (7). $\mathbf{x}_{s+1}$ is mapped to noise of a quarter variance, while $\mathbf{y}_s$ is mapped to noise of $c$-factored variance (see fig. 1). Although this is optional, it preserves interpretation between the single- and multi-resolution models.

## 4 Related Work

Multi-resolution approaches already serve as a key component of state-of-the-art GAN (Denton et al., 2015; Karras et al., 2018; Karnewar & Wang, 2020) and VAE (Razavi et al., 2019; Vahdat & Kautz, 2020) based deep generative models. Deconvolutional CNNs (Long et al., 2015; Radford et al., 2015) use upsampling layers to generate images more effectively. Modern state-of-the-art generative models have also injected noise at different levels to improve sample quality (Brock et al., 2019; Karras et al., 2020; Vahdat & Kautz, 2020).

Several prior works on normalizing flows (Kingma & Dhariwal, 2018; Hoogeboom et al., 2019a;b; Song et al., 2019; Ma et al., 2019; Durkan et al., 2019; Chen et al., 2020; Ho et al., 2019a; Lee et al., 2020; Yu et al., 2020) build on RealNVP (Dinh et al., 2017). Although they achieve great results in terms of BPD and image quality, they nonetheless report results from significantly higher number of parameters (some with 100x!), and several times GPU hours of training. However, many prior works don't report these metrics.

STEER (Ghosh et al., 2020) introduced temporal regularization to CNFs by making the final time of integration stochastic. However, we found that this increased training time without significant BPD improvement.

**Comparison to WaveletFlow**: We emphasize that there are important and crucial differences between our MRCNF and WaveletFlow. We generalize the notion of a multi-resolution image representation (section 3.2), and show that Wavelets are one case of this general formulation. WaveletFlow builds on the Glow (Kingma & Dhariwal, 2018) architecture, while ours builds on CNFs (Grathwohl et al., 2019; Finlay et al., 2020). We also make use of the notion of multi-resolution decomposition of the noise, which is optional, but is not taken into account by WaveletFlow. WaveletFlow uses an orthogonal transformation which does not preserve range ; our MRCNF uses eq. (7) which is volume-preserving and range-preserving. Finally, WaveletFlow applies special sampling techniques to obtain better samples from its model. We have so far not used such techniques for generation, but we believe they can potentially help our models as well. By making these important changes, we fix many of the previously discussed issues with WaveletFlow. For a more detailed ablation study, please check subsection 5.1.

**"Multiple scales" in prior normalizing flows**: Normalizing flows (Dinh et al., 2017; Kingma & Dhariwal, 2018; Grathwohl et al., 2019) try to be "multi-scale" by transforming the input in a smart way (squeezing operation) such that the width of the features progressively reduces in the direction of image to noise, while maintaining the total dimensions. This happens while operating at a *single resolution*. In contrast, our model stacks normalizing flows at *multiple resolutions* in an autoregressive fashion by conditioning on the images at coarser resolutions.

## 5 EXPERIMENTAL RESULTS

We train MRCNF models on the CIFAR10 (Krizhevsky et al., 2009) dataset at finest resolution of 32x32, and the ImageNet (Deng et al., 2009) dataset at 32x32, 64x64, 128x128. We build on top of the code provided in Finlay et al. (2020)[1]. Our code is attached in the supplementary material. In all cases, we train using *only one* NVIDIA RTX 20280 Ti GPU with 11GB.

In Table 1, we compare our results with prior work in terms of (lower is better in all cases) the BPD of the images of the test datasets under the trained models, the number of parameters used by the model, and the number of GPU hours taken to train. The most relevant models for comparison are the 1-resolution FFJORD (Grathwohl et al., 2019) models, and their regularized version RNODE (Finlay et al., 2020), since our model directly converts their architecture into multi-resolution. Other relevant comparisons are previous flow-based methods (Dinh et al., 2017; Kingma & Dhariwal, 2018; Song et al., 2019; Ho et al., 2019a; Yu et al., 2020), however their core architecture (RealNVP (Dinh et al., 2017)) is quite different from FFJORD.

**BPD**: At lower resolution spaces, we achieve comparable BPDs in lesser time with far fewer parameters than previous normalizing flows (and non flow-based approaches). However, the power of the multi-resolution formulation is more evident at higher resolutions: we achieve better BPD for ImageNet64 and ImageNet128 (Table 2) with significantly fewer parameters and lower time using only one GPU.

It is to be noted that we were not able to reproduce the same BPD as provided by STEER (Ghosh et al., 2020), we report the results of our re-implementation in Table 1. A more complete table can be found in the appendix. We also include analysis of out-of-distribution properties in the appendix.

---

[1]https://github.com/cfinlay/ffjord-rnode

Table 1: Bits-per-dimension (lower is better) of images in the corresponding evaluation sets for CIFAR10, ImageNet 32×32, and ImageNet 64×64. We also report the number of parameters in the models, and the time taken to train (in GPU hours). All our models were trained on only one GPU.
  Blank spaces indicate unreported values.    ‡As reported in Ghosh et al. (2020).    §Re-implemented by us.
‘x’: Fails to train.    *RNODE (Finlay et al., 2020) used 4 GPUs to train on ImageNet64.

| | CIFAR10 | | | IMAGENET32 | | | IMAGENET64 | | |
|---|---|---|---|---|---|---|---|---|---|
| | BPD | PARAM | TIME | BPD | PARAM | TIME | BPD | PARAM | TIME |
| **Non Flow-based Prior Work** | | | | | | | | | |
| Gated PixelCNN (Van den Oord et al., 2016) | 3.03 | | | 3.83 | | 60 | 3.57 | | 60 |
| SPN (Menick & Kalchbrenner, 2019) | | | | 3.85 | 150.0M | | 3.53 | 150.0M | |
| Sparse Transformer (Child et al., 2019) | 2.80 | 59.0M | | | | | 3.44 | 152.0M | 7days |
| NVAE (Vahdat & Kautz, 2020) | 2.91 | | 55 | 3.92 | | 70 | | | |
| DistAug (Jun et al., 2020) | 2.56 | 152.0M | | | | | 3.42 | 152.0M | |
| **Flow-based Prior Work** | | | | | | | | | |
| RealNVP (Dinh et al., 2017) | 3.49 | | | 4.28 | 46.0M | | 3.98 | 96.0M | |
| Glow (Kingma & Dhariwal, 2018) | 3.35 | 44.0M | | 4.09 | 66.1M | | 3.81 | 111.1M | |
| MaCow (Ma et al., 2019) | 3.16 | 43.5M | | | | | 3.69 | 122.5M | |
| Flow++ (Ho et al., 2019a) | 3.08 | 31.4M | | 3.86 | 169.0M | | 3.69 | 73.5M | |
| Wavelet Flow (Yu et al., 2020) | | | | 4.08 | 64.0M | | 3.78 | 96.0M | 822 |
| DenseFlow (Grcić et al., 2021) | 2.98 | | 250 | 3.63 | | 310 | 3.35 | | 224 |
| **1-Resolution Continuous Normalizing Flow** | | | | | | | | | |
| FFJORD (Grathwohl et al., 2019) | 3.40 | 0.9M | ≥5days | ‡3.96 | ‡2.0M | ‡>5days | x | | x |
| RNODE (Finlay et al., 2020) | 3.38 | 1.4M | 31.8 | ‡2.36 §3.49 | 2.0M §1.6M | ‡30.1 §40.4 | *3.83 | 2.0M | *256.4 |
| FFJORD + STEER (Ghosh et al., 2020) | 3.40 | 1.4M | 86.3 | 3.84 | 2.0M | >5days | | | |
| RNODE + STEER (Ghosh et al., 2020) | 3.397 | 1.4M | 22.2 | 2.35 §3.49 | 2.0M §1.6M | 24.9 §30.1 | | | |
| **(OURS) Multi-Resolution Continuous Normalizing Flow (MRCNF)** | | | | | | | | | |
| 2-resolution MRCNF (small) | 3.65 | 1.3M | 19.8 | 3.77 | 1.3M | 18.2 | 3.44 | 2.0M | 42.3 |
| 2-resolution MRCNF (big) | 3.54 | 3.3M | 36.5 | 3.78 | 6.7M | 18.0 | x | 6.7M | x |
| 3-resolution MRCNF (small) | 3.79 | 1.5M | 17.4 | 3.97 | 1.5M | 13.8 | 3.55 | 2.0M | 35.4 |
| 3-resolution MRCNF (big) | 3.60 | 5.1M | 38.3 | 3.93 | 10.2M | 41.2 | x | 7.6M | x |

**Train time**: All our experiments used only one GPU, and took significantly less time to train than 1-resolution CNFs, and all prior works including flow-based and non-flow-based models. Since all the CNFs can be trained in parallel, the actual training time in practice could be much lower than reported.

**Super-resolution**: Our formulation also allows for super-resolution of images (Figure 3) free of cost since our framework is autoregressive in resolution.

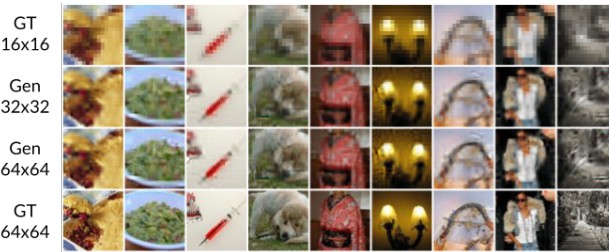

Figure 3: Examples of super-resolving from ImageNet 16×16 to 64×64. GT: ground truth, Gen: generated.

**Progressive training**: We trained an MRCNF model on ImageNet128 by training only the finest resolution (128×128) conditioned on the immediate coarser (64×64) images, and attached it to a 3-resolution model trained on ImageNet64. The resultant 4-resolution ImageNet128 model gives a BPD of 3.31 (Table 2) with just 2.74M parameters in ≈60 GPU hours.

Table 2: Metrics for unconditional ImageNet128 generation.

| IMAGENET128 | BPD | PARAM | TIME |
|---|---|---|---|
| Parallel Multiscale (Reed et al., 2017) | 3.55 | | |
| SPN (Menick & Kalchbrenner, 2019) | 3.08 | 250.00M | |
| **(OURS) 4-resolution MRCNF** | 3.31 | 2.74M | 58.59 |

## 5.1 ABLATION STUDY

Our MRCNF method differs from WaveletFlow in three respects: 1. we use CNFs, 2. we use eq. (7) instead of eq. (6) as used by WaveletFlow, 3. we use multi-resolution noise. We check the individual effects of these changes in an ablation study in Table 3, and conclude that:

Table 3: Ablation study across using Wavelet in eq. (6), and multi-resolution noise formulation in 3.4.

|  | CIFAR10 | | | IMAGENET64 | | |
|---|---|---|---|---|---|---|
|  | BPD | PARAM | TIME | BPD | PARAM | TIME |
| WaveletFlow (Yu et al., 2020) |  |  |  | 3.78 | 98.0M | 822.00 |
| 1-resolution CNF (RNODE) (Finlay et al., 2020) | 3.38 | 1.4M | 31.84 | 3.83 | 2.0M | 256.40 |
| **2-resolution** | | | | | | |
| eq. (6) WaveletFlow, but with CNFs, w/o multi-res noise | 3.68 | 1.3M | 27.25 | x | 2.0M | x |
| eq. (6) WaveletFlow, but with CNFs, w/ multi-res noise | 3.69 | 1.3M | 25.88 | x | 2.0M | x |
| eq. (7) MRCNF w/o multi-res noise | 3.66 | 1.3M | 19.79 | 3.48 | 2.0M | 42.33 |
| eq. (7) MRCNF w/ multi-res noise (**Ours**) | 3.65 | 1.3M | 19.69 | 3.44 | 2.0M | 42.30 |
| **3-resolution** | | | | | | |
| eq. (6) WaveletFlow, but with CNFs, w/o multi-res noise | 3.82 | 1.5M | 22.99 | 3.62 | 2.0M | 43.37 |
| eq. (6) WaveletFlow, but with CNFs, w/ multi-res noise | 3.82 | 1.5M | 25.28 | 3.62 | 2.0M | 44.21 |
| eq. (7) MRCNF w/o multi-res noise | 3.79 | 1.5M | 17.25 | 3.57 | 2.0M | 35.42 |
| eq. (7) MRCNF w/ multi-res noise (**Ours**) | 3.79 | 1.5M | 17.44 | 3.55 | 2.0M | 35.39 |

1. Simply replacing the normalizing flows in WaveletFlow with CNFs does not produce the best results. It does improve the BPD and training time compared to WaveletFlow.

2. Using our unimodular transformation in eq. (7) instead of the original Wavelet Transformation of eq. (6) not only improves the BPD, it also consistently decreases training time. Hence, although a model could learn to compensate for the additional BPD in eq. (6), we find that using eq. (7) makes training better and faster.

3. As expected, the use of multi-resolution noise does not have a critical impact on either BPD or training time. We use it anyway so as to retain interpretation with 1-resolution models.

Thus, our MRCNF model is not a trivial replacement of normalizing flows with CNFs in WaveletFlow. We generalize the notion of multi-resolution image representation, in which the Discrete Wavelet Transform is one of many possibilities. We then derived a unimodular transformation that adds no change in likelihood.

## 6 CONCLUSION

We have presented a Multi-Resolution approach to Continuous Normalizing Flows (MRCNF). MRCNF models achieve comparable or better performance in significantly less training time, training on a single GPU, with a fraction of the number of parameters of other competitive models. We achieved this by designing a general formulation for a multi-resolution image representation, and derived a unimodular transformation that adds no cost in terms of log-likelihood. Although the likelihood values for 32×32 resolution datasets such as CIFAR10 and ImageNet32 do not improve significantly over the baseline, ImageNet64 and above see a marked improvement. The performance is better for higher resolutions, as seen in the case of ImageNet128.

We also conducted an ablation study to note the effects of each change introduced in the formulation. We found that our unimodular transformation has a significant contribution towards improving BPD and decreasing training time.

ETHICS STATEMENT

In terms of broader social impacts of this work, generative models of images can be used to generate so-called fake images, and this issue has been discussed at length in other works. We have tried to stay away from particularly concerning use of our work. We have not performed experiments on face-specific datasets. Although the ImageNet dataset contains human face images, we have only tested our method on an unconditional model, and have made no effort towards controllable generation. An increasing concern in the machine learning community is that of increased use of computational resources and its impact on the environment. We emphasize lower computational budgets, and achieve comparable performance with far fewer parameters and significantly less training time.

REPRODUCIBILITY STATEMENT

To ensure reproducibility of our work, we have attached anonymized version of our code repository in the supplementary material. We have mentioned that our code builds on top of a public code repository of the most relevant prior method: https://github.com/cfinlay/ffjord-rnode. We have included the exact mathematical equations governing our method in Section 3.3, which can be verified in our code. We have mentioned other details of our models in the appendix : number of channels, number of blocks, corresponding total number of parameters. We have also mentioned in the appendix that we perform gradient clipping, and added relevant details there. Our code logs and plots the BPD, number of parameters, and training time for every experiment, and saves samples of generated images at regular intervals. It also plots other values of interest such as change in log likelihood, log probability of mapped noise samples, etc. The code also includes options for launching experiments with configurations of prior methods including FFJORD (Grathwohl et al., 2019), RNODE (Finlay et al., 2020), STEER (Ghosh et al., 2020), and WaveletFlow (Yu et al., 2020) using CNFs.

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

# APPENDIX

## A  OUT-OF-DISTRIBUTION PROPERTIES OF (MR)CNF

The derivation of likelihood-based models suggests that the density of an image under the model is an effective measure of its likelihood of being in distribution. However, recent works (Theis et al., 2016; Nalisnick et al., 2019a; Serrà et al., 2020; Nalisnick et al., 2019b) have pointed out that it is possible that images drawn from other distributions have higher model likelihood. Examples have been shown where normalizing flow models (such as Glow) trained on CIFAR10 images assign higher likelihood to SVHN (Netzer et al., 2011) images. This could have serious implications on the practical applicability of these models. Some also note that likelihood-based models do not generate images with good sample quality as they avoid assigning small probability to out-of-distribution (OoD) data points, hence using model likelihood (-BPD) for detecting OoD data is not effective.

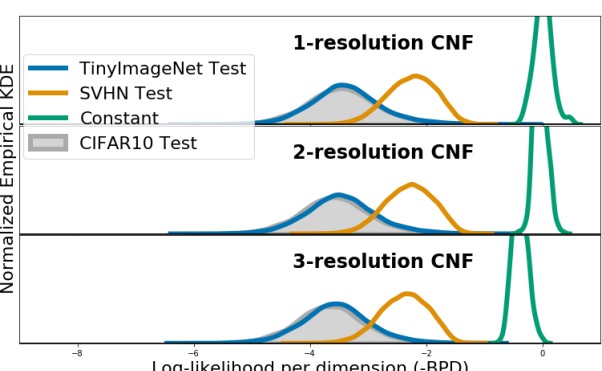

Figure 4: Histogram of log likelihood per dimension of out-of-distribution datasets (TinyImageNet, SVHN, Constant) under (MR)CNF models trained on CIFAR10. As with other likelihood-based generative models such as Glow & PixelCNN, OoD datasets have higher likelihood under (MR)CNFs.

We conduct the same experiments with (MR)CNFs, and find that similar conclusions can be drawn. Figure 4 plots the histogram of log likelihood per dimension (-BPD) of OoD images (SVHN, TinyImageNet) under MRCNF models trained on CIFAR10. It can be observed that the likelihood of the OoD SVHN is higher than CIFAR10 for MRCNF, similar to the findings for Glow, PixelCNN, VAE in earlier works (Nalisnick et al., 2019a; Choi et al., 2018; Serrà et al., 2020; Nalisnick et al., 2019b; Kirichenko et al., 2020).

One possible explanation put forward by Nalisnick et al. (2019b) is that "typical" images are less "likely" than constant images, which is a consequence of the distribution of a Gaussian in high dimensions. Indeed, as our Figure 4 shows, constant images have the highest likelihood under MRCNFs, while randomly generated (uniformly distributed) pixels have the least likelihood (not shown in figure due to space constraints).

Choi et al. (2018); Nalisnick et al. (2019b) suggest using "typicality" as a better measure of OoD. However, Serrà et al. (2020) observe that the complexity of an image plays a significant role in the training of likelihood-based generative models. They propose a new metric $S$ as an out-of-distribution detector:

$$S(\mathbf{x}) = \mathrm{bpd}(\mathbf{x}) - L(\mathbf{x}) \tag{16}$$

where $L(\mathbf{x})$ is the complexity of an image $\mathbf{x}$ measured as the length of the best compressed version of $\mathbf{x}$ (we use FLIF (Sneyers & Wuille, 2016) following Serrà et al. (2020)) normalized by the number of dimensions.

We perform a similar analysis as Serrà et al. (2020) to test how $S$ compares with -bpd for OoD detection. For different MRCNF models trained on CIFAR10, we compute the area under the receiver operating characteristic curve (auROC) using -bpd and $S$ as standard evaluation for the OoD detection task (Hendrycks et al., 2019; Serrà et al., 2020).

Table 4 shows that $S$ does perform better than -bpd in the case of (MR)CNFs, similar to the findings in Serrà et al. (2020) for

Table 4: auROC for OoD detection using -bpd and $S$ (Serrà et al., 2020), for models trained on CIFAR10.

| CIFAR10 | SVHN | | TIN | |
|---|---|---|---|---|
| (trained) | -bpd | S | -bpd | S |
| Glow | 0.08 | 0.95 | 0.66 | 0.72 |
| 1-res CNF | 0.07 | 0.16 | 0.48 | 0.60 |
| 2-res MRCNF | 0.06 | 0.25 | 0.46 | 0.66 |
| 3-res MRCNF | 0.05 | 0.25 | 0.46 | 0.66 |

Glow and PixelCNN++. It seems that SVHN is easier to detect as OoD for Glow than MRCNFs. However, OoD detection performance is about the same for TinyImageNet. We also observe that MRCNFs are better at OoD than CNFs.

Other OoD methods (Hendrycks & Gimpel, 2017; Liang et al., 2018; Lee et al., 2018; Sabeti & Høst-Madsen, 2019; Høst-Madsen et al., 2019; Hendrycks et al., 2019) are not suitable in our case, as identified in Serrà et al. (2020).

### A.1 SHUFFLED IN-DISTRIBUTION IMAGES

Kirichenko et al. (2020) conclude that normalizing flows do not represent images based on their semantic contents, but rather directly encode their visual appearance. We verify this for continuous normalizing flows by estimating the density of in-distribution test images, but with patches of pixels randomly shuffled. Figure 5 (a) shows an example of images of shuffled patches of varying size, Figure 5 (b) shows the graph of the their log-likelihoods.

That shuffling pixel patches would render the image semantically meaningless is reflected in the Fréchet Inception Distance (FID) between CIFAR10-Train and these sets of shuffled images — 1x1: 340.42, 2x2: 299.99, 4x4: 235.22, 8x8: 101.36, 16x16: 33.06, 32x32 (i.e. CIFAR10-Test): 3.15. However, we see that images with large pixel patches shuffled are quite close in likelihood to the unshuffled images, suggesting that since their visual content has not changed much they are almost as likely as unshuffled images under MRCNFs.

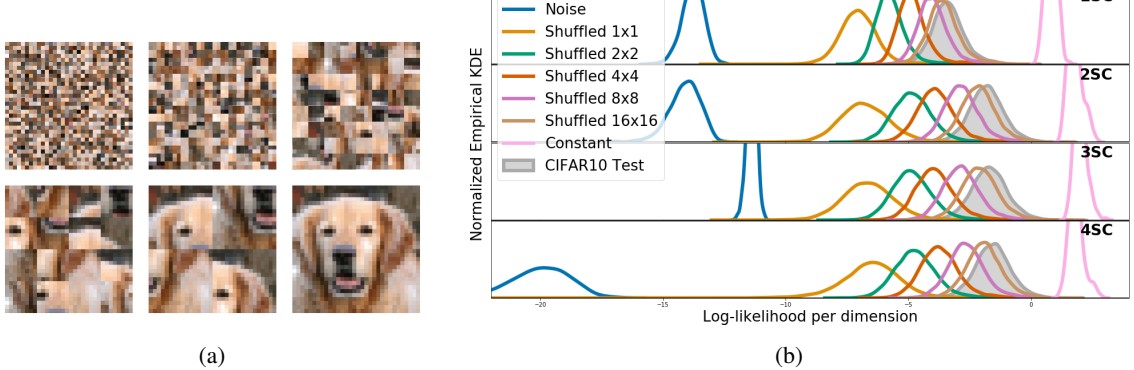

(a)          (b)

Figure 5: (a) Example of shuffling different-sized patches of a 32×32 image: (left to right, top to bottom) 1×1, 2×2, 4×4, 8×8, 16×16, 32×32 (unshuffled) (b) Bits-per-dim vs Epoch at each resolution for different MRCNF models trained on CIFAR10.

## B  FULL TABLE 1

Table 5: Unconditional image generation metrics (lower is better in all cases): number of parameters in the model, bits-per-dimension, time (in hours). Most previous models use multiple GPUs for training, all our models were trained on only *one* NVIDIA V100 GPU. [‡]As reported in Ghosh et al. (2020). [*]FFJORD RNODE Finlay et al. (2020) used 4 GPUs to train on ImageNet64. 'x': Fails to train.

| | CIFAR10 | | | IMAGENET32 | | | IMAGENET64 | | |
|---|---|---|---|---|---|---|---|---|---|
| | BPD | PARAM | TIME | BPD | PARAM | TIME | BPD | PARAM | TIME |
| **Non Flow-based Prior Work** | | | | | | | | | |
| PixelRNN (Oord et al., 2016) | 3.00 | | | 3.86 | | | 3.63 | | |
| Gated PixelCNN (Van den Oord et al., 2016) | 3.03 | | | 3.83 | | 60 | 3.57 | | 60 |
| Parallel Multiscale (Reed et al., 2017) | | | | 3.95 | | | 3.70 | | |
| Image Transformer Parmar et al. (2018) | 2.90 | | | 3.77 | | | | | |
| PixelSNAIL (Chen et al., 2018b) | 2.85 | | | 3.80 | | | | | |
| SPN (Menick & Kalchbrenner, 2019) | | | | 3.85 | 150.0M | | 3.53 | 150.0M | |
| Sparse Transformer (Child et al., 2019) | 2.80 | 59.0M | | | | | 3.44 | 152.0M | 7days |
| Axial Transformer (Ho et al., 2019b) | | | | 3.76 | | | 3.44 | | |
| PixelFlow++ (Nielsen & Winther, 2020) | 2.92 | | | | | | | | |
| NVAE (Vahdat & Kautz, 2020) | 2.91 | | 55 | 3.92 | | 70 | | | |
| Dist-Aug Sparse Transformer (Jun et al., 2020) | 2.56 | 152.0M | | | | | 3.42 | 152.0M | |
| **Flow-based Prior Work** | | | | | | | | | |
| IAF (Kingma et al., 2016) | | | | 3.11 | | | | | |
| RealNVP (Dinh et al., 2017) | 3.49 | | | 4.28 | 46.0M | | 3.98 | 96.0M | |
| Glow (Kingma & Dhariwal, 2018) | 3.35 | 44.0M | | 4.09 | 66.1M | | 3.81 | 111.1M | |
| i-ResNets (Behrmann et al., 2019) | | | | | | | | | |
| Emerging (Hoogeboom et al., 2019a) | 3.34 | 44.7M | | 4.09 | 67.1M | | 3.81 | 67.1M | |
| IDF (Hoogeboom et al., 2019b) | 3.34 | | | 4.18 | | | 3.90 | | |
| S-CONF (Karami et al., 2019) | 3.34 | | | | | | | | |
| MintNet (Song et al., 2019) | 3.32 | 17.9M | ≥5days | 4.06 | 17.4M | | | | |
| Residual Flow (Chen et al., 2019) | 3.28 | | | 4.01 | | | 3.76 | | |
| MaCow (Ma et al., 2019) | 3.16 | 43.5M | | | | | 3.69 | 122.5M | |
| Neural Spline Flows (Durkan et al., 2019) | 3.38 | 11.8M | | | | | 3.82 | 15.6M | |
| Flow++ (Ho et al., 2019a) | 3.08 | 31.4M | | 3.86 | 169.0M | | 3.69 | 73.5M | |
| ANF (Huang et al., 2020) | 3.05 | | | 3.92 | | | 3.66 | | |
| MEF (Xiao & Liu, 2020) | 3.32 | 37.7M | | 4.05 | 37.7M | | 3.73 | 46.6M | |
| VFlow (Chen et al., 2020) | 2.98 | | | 3.83 | | | | | |
| Woodbury NF (Lu & Huang, 2020) | 3.47 | | | 4.20 | | | 3.87 | | |
| NanoFlow (Lee et al., 2020) | 3.25 | | | | | | | | |
| ConvExp (Hoogeboom et al., 2020) | 3.218 | | | | | | | | |
| Wavelet Flow (Yu et al., 2020) | | | | 4.08 | 64.0M | | 3.78 | 96.0M | 822 |
| TayNODE (Kelly et al., 2020) | 1.039 | | | | | | | | |
| **1-resolution Continuous Normalizing Flow** | | | | | | | | | |
| FFJORD (Grathwohl et al., 2019) | 3.40 | 0.9M | ≥5days | [‡]3.96 | [‡]2.0M | [‡]>5days | x | | x |
| RNODE (Finlay et al., 2020) | 3.38 | 1.4M | 31.84 | [‡]2.36 [§]3.49 | 2.0M [§]1.6M | [‡]30.1 [§]40.39 | [*]3.83 | 2.0M | [*]256.4 |
| FFJORD + STEER (Ghosh et al., 2020) | 3.40 | 1.4M | 86.34 | 3.84 | 2.0M | >5days | | | |
| RNODE + STEER (Ghosh et al., 2020) | 3.397 | 1.4M | 22.24 | 2.35 [§]3.49 | 2.0M [§]1.6M | 24.90 [§]30.07 | | | |
| **(OURS) Multi-Resolution Continuous Normalizing Flow (MRCNF)** | | | | | | | | | |
| 2-resolution MRCNF | 3.65 | 1.3M | 19.79 | 3.77 | 1.3M | 18.18 | 3.44 | 2.0M | 42.30 |
| 2-resolution MRCNF | 3.54 | 3.3M | 36.47 | 3.78 | 6.7M | 17.98 | x | 6.7M | x |
| 3-resolution MRCNF | 3.79 | 1.5M | 17.44 | 3.97 | 1.5M | 13.78 | 3.55 | 2.0M | 35.39 |
| 3-resolution MRCNF | 3.60 | 5.1M | 38.27 | 3.93 | 10.2M | 41.20 | x | 7.6M | x |

## C   QUALITATIVE SAMPLES

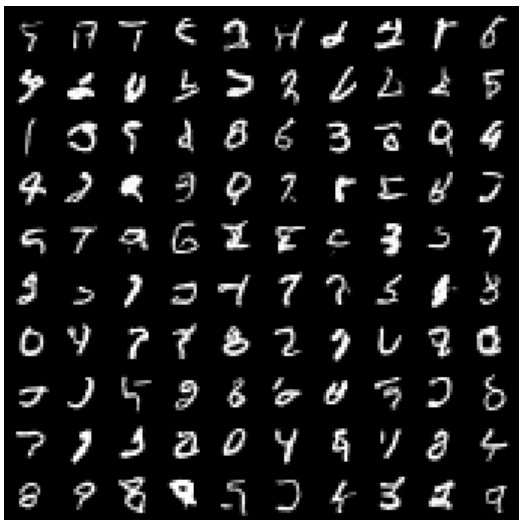

(a) Generated samples at 16×16                    (b) Corresponding generated samples at 32×32

Figure 6: Generated samples from MNIST.

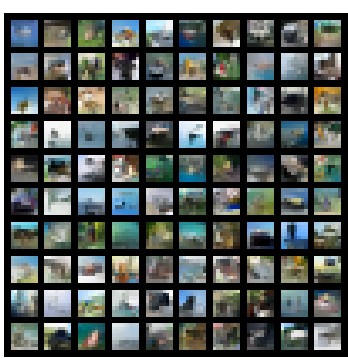
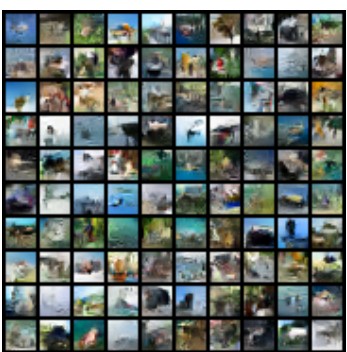
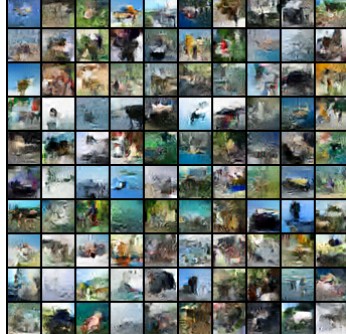

(a) Generated samples at 8×8          (b) Generated samples at 16×16          (c) Generated samples at 32×32

Figure 7: Generated samples from CIFAR10.

## D   SIMPLE EXAMPLE OF DENSITY ESTIMATION

For example, if we use Euler method as our ODE solver, for density estimation Equation 2 reduces to:

$$\mathbf{v}(t_1) = \mathbf{v}(t_0) + (t_1 - t_0)f_s(\mathbf{v}(t_0), t_0 \mid \mathbf{c}) \tag{17}$$

where $f_s$ is a neural network, $t_0$ represents the "time" at which the state is image $\mathbf{x}$, and $t_1$ is when the state is noise $\mathbf{z}$. We start at scale S with an image sample $\mathbf{x}_S$, and assume $t_0$ and $t_1$ are 0 and 1 respectively:

$$
\begin{cases}
\mathbf{z}_S = \mathbf{x}_S + f_S(\mathbf{x}_S,\ t_0 \mid \mathbf{x}_{S-1}) \\
\mathbf{z}_{S-1} = \mathbf{x}_{S-1} + f_{S-1}(\mathbf{x}_{S-1},\ t_0 \mid \mathbf{x}_{S-2}) \\
\ \vdots \\
\mathbf{z}_1 = \mathbf{x}_1 + f_1(\mathbf{x}_1,\ t_0 \mid \mathbf{x}_0) \\
\mathbf{z}_0 = \mathbf{x}_0 + f_0(\mathbf{x}_0,\ t_0)
\end{cases}
\tag{18}
$$

## E    SIMPLE EXAMPLE OF GENERATION

For example, if we use Euler method as our ODE solver, for generation Equation 2 reduces to:

$$
\mathbf{v}(t_0) = \mathbf{v}(t_1) + (t_0 - t_1) f_s(\mathbf{v}(t_1), t_1 \mid \mathbf{c})
\tag{19}
$$

i.e. the state is integrated backwards from $t_1$ (i.e. $\mathbf{z}_s$) to $t_0$ (i.e. $\mathbf{x}_s$). We start at scale 0 with a noise sample $\mathbf{z}_0$, and assume $t_0$ and $t_1$ are 0 and 1 respectively:

$$
\begin{cases}
\mathbf{x}_0 = \mathbf{z}_0 - f_0(\mathbf{z}_0,\ t_1) \\
\mathbf{x}_1 = \mathbf{z}_1 - f_1(\mathbf{z}_1,\ t_1 \mid \mathbf{x}_0) \\
\ \vdots \\
\mathbf{x}_{S-1} = \mathbf{z}_{S-1} - f_{S-1}(\mathbf{z}_{S-1},\ t_1 \mid \mathbf{x}_{S-2}) \\
\mathbf{x}_S = \mathbf{z}_S - f_S(\mathbf{z}_S,\ t_1 \mid \mathbf{x}_{S-1})
\end{cases}
\tag{20}
$$

## F    MODELS

We used the same neural network architecture as in RNODE (Finlay et al., 2020). The CNF at each resolution consists of a stack of $bl$ blocks of a 4-layer deep convolutional network comprised of 3x3 kernels and softplus activation functions, with 64 hidden dimensions, and time t concatenated to the spatial input. In addition, except at the coarsest resolution, the immediate coarser image is also concatenated with the state. The integration time of each piece is [0, 1]. The number of blocks $bl$ and the corresponding total number of parameters are given in Table 6.

Table 6: Number of parameters for different models with different total number of resolutions (res), and the number of channels (ch) and number of blocks (bl) per resolution.

| MRCNF | | | |
|---|---|---|---|
| resolutions | ch | bl | Param |
| | 64 | 2 | 0.16M |
| 1 | 64 | 4 | 0.32M |
| | 64 | 14 | 1.10M |
| | 64 | 8 | 1.33M |
| 2 | 64 | 20 | 3.34M |
| | 64 | 40 | 6.68M |
| | 64 | 6 | 1.53M |
| 3 | 64 | 8 | 2.04M |
| | 64 | 20 | 5.10M |

## G   GRADIENT NORM

In order to avoid exploding gradients, We clipped the norm of the gradients (Pascanu et al., 2013) by a maximum value of 100.0. In case of using adversarial loss, we first clip the gradients provided by the adversarial loss by 50.0, sum up the gradients provided by the log-likelihood loss, and then clip the summed gradients by 100.0.

## H   8-BIT TO UNIFORM

The change-of-variables formula gives the change in probability due to the transformation of $\mathbf{u}$ to $\mathbf{v}$:

$$\log p(\mathbf{u}) = \log p(\mathbf{v}) + \log \left| \det \frac{d\mathbf{v}}{d\mathbf{u}} \right|$$

Specifically, the change of variables from an 8-bit image to an image with pixel values in range [0, 1] is:

$$\mathbf{b}_S^{(p)} = \frac{\mathbf{a}_S^{(p)}}{256}$$

$$\implies \log p(\mathbf{a}_S) = \log p(\mathbf{b}_S) + \log \left| \det \frac{d\mathbf{b}}{d\mathbf{a}} \right|$$

$$\implies \log p(\mathbf{a}_S) = \log p(\mathbf{b}_S) + \log \left( \frac{1}{256} \right)^{D_S}$$

$$\implies \log p(\mathbf{a}_S) = \log p(\mathbf{b}_S) - D_S \log 256$$

$$\implies \mathrm{bpd}(\mathbf{a}_S) = \frac{-\log p(\mathbf{a}_S)}{D_S \log 2}$$

$$= \frac{-(\log p(\mathbf{b}_S) - D_S \log 256)}{D_S \log 2}$$

$$= \frac{-\log p(\mathbf{b}_S)}{D_S \log 2} + \frac{\log 256}{\log 2}$$

$$= \mathrm{bpd}(\mathbf{x}) + 8$$

where $\mathrm{bpd}(\mathbf{x})$ is given from Equation 13.

## I   FID V/S TEMPERATURE

Table 7 lists the FID values of generated images from MRCNF models trained on CIFAR10, with different temperature settings on the Gaussian.

Table 7: FID v/s temperature for MRCNF models trained on CIFAR10.

| | Temperature | | | | | |
|---|---|---|---|---|---|---|
| | 1.0 | 0.9 | 0.8 | 0.7 | 0.6 | 0.5 |
| **1-resolution CNF** | 138.82 | 147.62 | 175.93 | 284.75 | 405.34 | 466.16 |
| **2-resolution MRCNF** | 89.55 | 106.21 | 171.53 | 261.64 | 370.38 | 435.17 |
| **3-resolution MRCNF** | 88.51 | 104.39 | 152.82 | 232.53 | 301.89 | 329.12 |

