# OpenReview forum: "Multi-Resolution Continuous Normalizing Flows"
_ICLR.cc/2022/Conference — ICLR 2022 Submitted_

### Official Review · Reviewer_McBP · 2021-11-01

**Correctness:** 3
**Technical Novelty And Significance:** 4
**Empirical Novelty And Significance:** 4
**Recommendation:** 6
**Confidence:** 4

**Main Review:**

The motivation and execution of the paper are good and I appreciate the many experiments and ablations the authors have run. However, I have a few concerns about the discussion of some of the results in the paper as well as questions around some aspects of the model. The main strengths and weaknesses (in my eyes) are described below.

**Strengths**:
- The paper is well motivated and the ideas are sound - it makes sense to take advantage of the multi resolution structure of images.
- The paper is well written and the figures are nice.
- The authors make extensive ablation studies and compare their model to a large number of baselines (including reimplementations of baselines which is great). I appreciate the effort the authors have put in, in order to have thorough experiments on a variety of image datasets.
- The results on Imagenet64 are impressive, getting low bpd with a low number of parameters and reduced training time.
- The progressive training results on Imagenet128 are very nice. The authors are able to achieve fairly good results by only training a single refining step on top of an Imagenet64 model, which is impressive.
- While the authors do not perform any experiments on this aspect of the model, the fact that the model can be trained in parallel is interesting.
- Discussion of related work is thorough and the model is well situated in the literature.
- The OoD experiments in the appendix are quite interesting.

**Weaknesses**:
- Some of the experimental results are slightly confusing. From both Table 1 and Table 3, it appears that adding more levels of resolutions (and hence more parameters) consistently _decreases_ performance, which seems very counterintuitive with the main point of the paper (that modeling multiple resolutions is important). This could suggest that it’s not the multi resolution aspect of the model that helps, but something else. I was disappointed that the authors do not seem to discuss this at all in the paper (in fact I had not realised this until I looked at the numbers in detail). This should be explicitly mentioned and discussed in the paper.
- Experimental results on low resolution datasets are not very good. For example, on CIFAR10, for all tested resolutions (2 and 3) the model does worse than the RNODE and vanilla FFJORD baselines. Again this does not seem to be discussed or acknowledged anywhere in the paper. It is not a problem as such that this doesn’t work well for low res datasets given that it works well for high res datasets, but it would be good to discuss this and be more explicit about limitations in general.

**Questions**:
- I am confused about how the conditioning on the coarse image is done exactly. This feels like quite a significant part of the model but as far as I can tell is only described by a single sentence on page 6. How is the input image of the CNF concatenated with the coarser image? The dimension of the input image is equal to the dimension of the coarser image + the dimension of the noise variables if I understood correctly, so how can the input image and the coarse image be concatenated (they have different shapes)? Shouldn’t the noise and the coarse image be concatenated instead? It would be good to be more explicit about how this works.
- In section 3.2, you choose $\mathbf{y}_1$ such that it “contains information not present in $\mathbf{x}_2$ such that $\mathbf{x}_1$ is obtained when $\mathbf{y}_1$ and $\mathbf{x}_2$ are combined”. A very simple thing to do here that would satisfy this criterion is to set $\mathbf{y}_1 = (x_1, x_2, x_3)$. Have you run this ablation?

**Typos**:
- Page 6: “we use to two regularization terms” -> “we use two regularization terms”


**Summary Of The Paper:**

This paper proposes a new architecture for continuous normalizing flows that explicitly models images at multiple resolutions. The authors achieve this by learning an unconditional distribution of coarse images and then learning conditional distributions at progressively finer resolutions, each with a continuous normalizing flow. In contrast to the typical “squeeze” layers used in normalizing flows, where the noise is only passed at the first layer, the proposed model injects noise at several resolutions (which has already proved successful with other generative models). The authors introduce a new (invertible) transformation to go from a coarse image to a finer image which preserves the range of the image while having unit determinant (implying the transformation leaves the log likelihood unchanged). The whole model is then trained via maximum likelihood.

The authors evaluate their method on standard image datasets and perform several ablations to test the contributions of their model.

The main contributions of the paper in my eyes are then:
- The introduction of a new architecture and layer for learning multi resolution normalizing flows
- Demonstration that the proposed model can be trained with limited computational resources even on fairly large scale datasets (such as Imagenet128)
- Ablation studies evaluating the improvements from each aspect of the proposed model


**Summary Of The Review:**

This paper introduces a new architecture for multi resolution continuous normalizing flows. The paper is well written and the model is sound and the authors achieve impressive results on Imagenet at high resolutions. However, as described in the weakness sections I believe there are still some aspects of the model and experiment discussion that need to be clarified. I therefore believe this paper is currently marginally above the acceptance threshold.

---

### Official Review · Reviewer_efZN · 2021-11-01

**Correctness:** 3
**Technical Novelty And Significance:** 2
**Empirical Novelty And Significance:** 2
**Recommendation:** 5
**Confidence:** 3

**Details Of Ethics Concerns:**

I have communicated my concern to the AC.

**Main Review:**

Strengths:
-The authors provide a faster multi-resolution strategy for normalizing flows.
-The empirical results are better than the benchmarked methods, e.g. WaveletFlow [1], Glow [2], FFJORD [3].
-The paper is clearly written and easy to read.
-Authors provide detailed descriptions and the source code thus the work should be easier to reproduce.

Weaknesses:
-Comparisons to some of the benchmarked methods are not fair/relevant (details below).
-Being faster to train, one would expect the method to be scaled up to larger images but no such experiments were reported. Thus, it is hard to evaluate how significant the training speedup is since the ODE based solutions are bottlenecked by training time [3], i.e. if the method does not address/alleviate the scalability issues, training faster on 64x64 images is less impactful.
-Clarifications are needed for various parts of the submission (details below).

Required clarifications:
-Section 3.2, the sentence: “This can be scaled up to larger spatial regions by performing the same calculation for each 2×2 patch.” I do not immediately see how. If “each” means a sliding window with stride 2, authors should explicitly write this. Otherwise some clarification is needed.
-Section 3.2: Why does the logdet term being zero makes the training faster? More generally, the authors propose a new M matrix but other than looking nice, the value of the novelty is not clear (i.e., why should an additive log term be problematic for the training speed?). Maybe the authors can stress the training speedup a bit more and provide some intuition about the cause of this speedup.
-Section 3.3, sentence with “...are generated conditioned on the entire coarser image...”: I do not see how. M is local, sliding it in stages (I assume) will still yield a (larger) local mapping. Could the authors expand their explanation? This is likely going to result in inconsistent textures, similar to WaveletFlow [1].
-Section 4, “Comparison to WaveletFlow” part: It is not feasible to ask the authors to apply their innovations on top of a Glow [2] based architecture. However, as it stands, their comparison to WaveletFlow [1] does not seem fair. The ablations section does not seem relevant either (more on that below). The authors might want to add a disclaimer here that core architectures, training strategy, hardware/software are extremely different. If possible, I suggest the authors limit the qualitative comparisons to FFJORD [3] and leave the WaveletFlow comparisons to the empirical results section only. This would also inform the reader better since I believe this paper is a direct extension of FFJORD [3] with a generalized wavelet multi-resolution strategy. The other perspective of “applying CNF etc. to WaveletFlow [1]” would be misleading.
-Table 3 and more generally Section 5.1: As I understand it, the comparisons are not between a WaveletFlow [1] modified to incorporate CNF vs. MRCNF but rather between MRCNF with Haar vs. MRCNF with the authors’ M matrix. This needs to be extremely clear.
Furthermore, bullet point (1) is confusing to me. Do the authors replace the flow steps in WaveletFlow with a CNF? If so, what were the exact architectural changes?
Finally, the last paragraph may be misleading (or at least unclear). The authors seem not to have modified (or even reproduced) WaveletFlow themselves, seeing as they report the exact same empirical results from the paper [1] and no mention of it in the provided code. Thus, the claim of  “MRCNF being more than WaveletFlow with CNF” is confusing - the experiments are just not there for proving/disproving this claim. I may be missing something, I ask the authors to please clarify.
-Supplementary: Authors mention that gradient clipping for the adversarial loss being different. I could not find another mention of the adversarial loss in the paper.

Minor points:
-Introduction: I would urge the authors to abstain from blanket statements like “... notoriously difficult to train…”. With the newer normalization, loss function and architectural innovations GANs, at least in my experience, are easier to train.
-Section 3.2, first paragraph: I understand that this is probably due to the page limit but if the authors can find the space, each bullet point should start as a new line.
-Figure 2: I believe this figure can be cut without impacting the readability of the paper.


[1] Jason Yu, Konstantinos Derpanis, and Marcus Brubaker. Wavelet flow: Fast training of high resolution normalizing flows. In Advances in Neural Information Processing Systems, 2020

[2] Durk P Kingma and Prafulla Dhariwal. Glow: Generative flow with invertible 1x1 convolutions. In Advances in neural information processing systems, pp. 10215–10224, 2018

[3] Will Grathwohl, Ricky T. Q. Chen, Jesse Bettencourt, Ilya Sutskever, and David Duvenaud. Ffjord: Free-form continuous dynamics for scalable reversible generative models. International Conference on Learning Representations, 2019


**Summary Of The Paper:**

In this work, the authors propose a multi-resolution strategy for continuous normalizing flows. The proposed approach consists of a general wavelet based decomposition/downsampling where the transformation obeys some useful mathematical properties such as volume and range preservation. Empirical results indicate likelihood estimation performance better or on par with benchmarked methods. The proposed model also trains significantly faster with fewer parameters.

**Summary Of The Review:**

As the authors also point out, using multi-resolution strategies is not novel in itself. Additionally, the reported experiments indicate that the proposed method is not mature enough to be applied to larger images, limiting its usefulness. However, despite the incremental novelty of the approach and limited empirical results I believe the authors propose a method with potential to be widely adopted in the normalizing flows models. I would be willing to update my recommendation based on the authors’ response to my clarification requests listed above.

---

### Official Review · Reviewer_cSSK · 2021-11-02

**Correctness:** 2
**Technical Novelty And Significance:** 2
**Empirical Novelty And Significance:** 2
**Recommendation:** 3
**Confidence:** 4

**Main Review:**

Overall, my general take is that the paper is relatively weak on both conceptual novelty as well as application performance. My main findings are below.

Cons:

Conceptual Novelty:
Though the paper tried to differentiate from prior work (WaveletFlow) in various aspects (using CNF rather than RealNVP, utilizing a volume and range preserving formulation), the main idea of multi-resolution conditional modeling of images at the next resolution based on the previous, is very much the same. In fact, I don't think the authors did a good job at justifying the few differentiating design factors from WaveletFlow. For instance, the authors claimed that the volume-preserving and range-preserving transformations are a key novelty but the effects of this transformation (as opposed to the wavelet transforms) are not studies in an experiment.

Experiment Results:
From the experiment results (Table 1), it's not clear that (1) the multiresolution formulation is more efficient, or (2) the generative modeling quality is improved, both of which I think are central to the claims of the paper.
- First, the BPD metric seems to get worse with more resolutions, which seems contrary to the central story?
- Second, even though the authors claimed in the paper that the model is "significantly faster with on-par performance", the performance seems quite a bit worse than even very early methods like RealNVP. While the model size seems smaller, that seems more likely be due to the use of FFJORD/CNF, rather than the multi-resolution formulation, which is an existing architecture the authors adopted, rather than a contribution of this paper.
- Third, while model size seems to be a main selling point of the story, being "smaller/cheaper with lower performance" does not feel very compelling. Very likely than not, when the "large" models are reduced using simple methods like reducing the number of feature channels / number of stacked flows, they can slim down to the same size with comparable or even performance.

Pros:

I do like that the authors are quite describing the mathematical details of the proposed multi-resolution transformations (Sec. 3.1, 3.2) which are easy to follow and enjoyable to read.

**Summary Of The Paper:**

The authors proposed a multi-resolution version continuous normalizing flows with the claim of faster training time resulting in comparable performance. The authors note a few key differences with previous work (WaveletFlow) - using CNF instead of realnvp / glow based architectures, modeling noise at multi-resolution, and utilizing a volume and range preserving formulation,

**Summary Of The Review:**

Overall, I enjoyed reading the paper and I really like the mutliresolution idea that leverages conditional probability modeling, but I think the claimed main conceptual novelties are weak and not well supported by experimental results. Therefore I cannot recommend acceptance at ICLR.

---

### Official Review · Reviewer_K92k · 2021-11-02

**Correctness:** 3
**Technical Novelty And Significance:** 2
**Empirical Novelty And Significance:** 2
**Recommendation:** 5
**Confidence:** 4

**Main Review:**

Comments/ questions:
- The paper is written well and easy to understand.
- The contributions do not seem to be significant enough yet from either a theoretical or practical perspective. I would request the authors to investigate further and work with larger image sizes (256 x 256 Glow paper). Perhaps more insight into why flows are worse on OOD from a multiscale perspective would be a good addition. Can we check if the problem is worse or better at coarser scales? Are constant images consistently assigned higher likelihoods at all image scales? If so, can't we regularize for this? When shuffling images, again use the multi-scale ideas and check which scales are affected most and how that is related to the tile size used in shuffling. I think these would be interesting and insightful additions to the paper.
- The main advantage of the method seems to be training times and number of parameters in a network which is good but there are other papers out there now that claim similar advantages with injective flows (Trumpets, UAI '21 Kothari et al, M-flows, NeurIPS' 20 Cranmer et al).  Rather than being yet another paper in the same direction, I think the author's would benefit greatly from investigating their key ideas a bit more. For example, a more in-depth analysis of noise injection at various scales and how that affects image generation would be good to have. A simple analysis could show how posterior samples of $x_s$ vary given, say $y_s, y_{s-1}$  and $x_{s}$.



**Summary Of The Paper:**

The paper proposes a multi-resolution variant of continuous normalizing flows for images. They show empirical results for image sizes upto 64 x 64 where they seem to be better than regular continuous normalizing flows. The key proposed benefit seems to be in number of parameters and training times.

**Summary Of The Review:**

I think that the current draft of this work makes incremental contributions to the area of normalizing flows for image synthesis. While I completely agree with the core motivations of bringing multi-resolution structures and scale-separated noise injection to CNFs, I believe the authors can make a much more significant contribution by analyzing how their core motivations play out in their proposed method.

---

### Official Review · Reviewer_hXQn · 2021-11-02

**Correctness:** 2
**Technical Novelty And Significance:** 2
**Empirical Novelty And Significance:** 2
**Recommendation:** 3
**Confidence:** 3

**Main Review:**

Strengths

The problem considered in this paper is challenging and the approach to solve it (upscaling via conditional flows) is adequate to the problem. The improvements compared to WaveletFlow seems to work in practice what can be observed while analysing the quantitative results (BPD). It is also very beneficial for that work that the authors analyse also training time and complexity of the model during the experimental stage. The OOD analysis (unfortunately provided in Supplementary) is also interesting in terms of possible other applications of the model. It is also important to mention, that the code is provided in the suplement, so reproducing the results should be easy.

Weaknesses

My major concern is about the contribution of the paper. The first two points in introduction that point out the contribution are non-informative in my opinion. The first refers to the transformation that does not add the cost to the training objective. Why is it so important for the model to not modify the likelihood cost with that transformation? If the (log) cost is negative, it can reduce the likelihood value while testing.

The second contribution is simply the name of the model and should be removed. Even for the third contribution I have the problem of understanding what components of the model are responsible for time and model capacity reduction. In my opinion the contribution of the paper is based on the different base flow model (CNF) and using unimodular transformation in eq. (7).  This contribution is rather incremental with respect to the WaveletFlow. Using CNF seems to be a good direction but is not motivated taking into account that any conditional flow can be used instead. The new transformation function does not provide a significant improvement compared to WaveletFlow and is motivated by reduction of the cost of transformation not the quality of model itself.

The second concern is about the provided results. The Table 1 is incomplete, some results are missing for reference approaches, it is sparse, and therefore, difficult to follow and analyse. The authors do not provide the qualitative results for high-resolution images as in WaveletFlow - such empirical would be beneficial to show the gain of using the approach provided in the paper. The selection of BPD criterion for quantitative evaluation is ok, but the goal of multi-resolution generative model is to generate good-quality images for larger dimensions. Using some standard measures like FID or related and comparison to GAN models would be beneficial for this case.

The third concern is about the quality of the paper - the provided manuscript is difficult to follow, the contribution is not expressed well and the description of the method should be more clear. I do not understand, why it is important to optimize BPD  directly instead of likelihood? For me it is just the scaling issue that does not affect the training procedure. Some interesting parts like OOD analysis are moved to suplement.

**Summary Of The Paper:**

The paper describes the novel approach to generate high-resolution images in progressive manner. The approach is based on set of conditional flows that take the image generated from previous stage as conditioning factor and generate the image with the higher resolution. The quality of the method is compared to the reference approaches, mainly focusing on WaveletFlow as the most similar method.

**Summary Of The Review:**

Unfortunately, the paper is rather incremental with respect to WaveletFlow, the contribution is not expressed well and qualitative analysis is missing. I would like to encourage authors for more extensive empirical evaluation and highlighting the novelty of the approach. Therefore, my current recommendation is to reject this work.

---

### Decision · Program_Chairs · 2022-01-20

**Decision:**

Reject

**Comment:**

### Description

The paper enhances flow-based generative models by putting them into a coarse-to fine multi-resolution framework. The key technical challenge as I understand is designing up-scaling conditional flow modules. Since the operation needs to be invertible, the paper carefully designs what degrees of freedom need to be injected in addition to the low resolution image to compose a higher resolution one.

### Decision
The paper received 5 expert and rather detailed reviews. I have read and understood the paper and all reviews. Reviewers remark that the paper is well written, addresses a challenging problem. However reviews were in a consensus on that the contribution of the paper is marginal. The average score was 4.4. The authors did not respond to reviewers  and did not update the paper. There was no post-rebuttal discussion and or additional feedback from reviewers. Therefore, must reject.

### Comments
I have only minor comments on the writing and organization of the paper.
There are many self-repetitions in the text, restating what was already said above in same or very similar sentences. Some questions studied in appendices are not presented in the main papar.